# Improving Adversarial Transferability in MLLMs via Dynamic Vision-Language Alignment Attack

## Abstract

Multimodal Large Language Models (MLLMs), built upon LLMs, have recently gained attention for their capabilities in image recognition and understanding. However, while MLLMs are vulnerable to adversarial attacks, the transferability of these attacks across different models remains limited, especially under targeted attack setting. Existing methods primarily focus on vision-specific perturbations but struggle with the complex nature of vision-language modality alignment. In this work, we introduce the Dynamic Vision-Language Alignment (DynVLA) Attack, a novel approach that injects dynamic perturbations into the vision-language connector to enhance generalization across diverse vision-language alignment of different models. Our experimental results show that DynVLA significantly improves the transferability of adversarial examples across various MLLMs, including BLIP2, InstructBLIP, MiniGPT4, LLaVA, and closed-source models such as Gemini.

## 1 Introduction

Multimodal Large Language Models (MLLMs) (Liu et al., 2024; Li et al., 2023; Dai et al., 2024; Zhu et al., 2023; Chen et al., 2024; Bai et al., 2023; Team et al., keyword) built upon Large Language Models (LLMs) (Brown et al., 2020; Anil et al., 2023; Touvron et al., 2023a;b; Dubey et al., 2024; Chiang et al., 2023) have achieved great success in addressing intricate vision-language tasks, such as image captioning (Lin et al., 2014) and visual question answering (Goyal et al., 2017). By aligning visual and language modalities, these models excel in generating coherent language responses to visual input, demonstrating exceptional capabilities in both visual comprehension and language generation.

Despite the remarkable advancements in Multimodal Large Language Models (MLLMs), they remain susceptible to adversarial attacks (Szegedy et al., 2013; Goodfellow et al., 2014; Madry et al., 2017; Xie et al., 2019; Long et al., 2022; Wang et al., 2023a), where carefully designed inputs can deceive the models into producing incorrect or misleading outputs. In addition, recent works (Zhao et al., 2024; Dong et al., 2023; Cheng et al., 2024; Schaeffer et al., 2024) have shown that MLLMs can be misled by transferable adversarial examples (Gu et al., 2023; Tramèr et al., 2017; Papernot et al., 2016; Liu et al., 2016), where adversarial examples that are generated to fool one MLLM can also successfully deceive others. For example, Zhao et al. (2024) matches the visual representation of the adversarial input with the representation of the target image generated by the target text. Dong et al. (2023) utilize ensemble of a set of vision encoders when attack. Cheng et al. (2024) improve the transferability by typography-based input transformation. One of the critical challenges in this space is the limited transferability of these adversarial examples across different MLLMs, especially under targeted attack scenarios. We hypothesize that most prior work in transfer-based attacks has primarily focused on the visual components of MLLMs, such as visual representation matching (Zhao et al., 2024) and pixel-level augmentations (Cheng et al., 2024), without considering the diversity in vision-language modality alignment in MLLMs caused by the different base language models.

To this end, we propose **Dyn**amic **V**ision-**L**anguage **A**lignment (DynVLA) attack to dynamically perturb vision-language modality alignment in MLLMs. In MLLMs, the alignment between vision

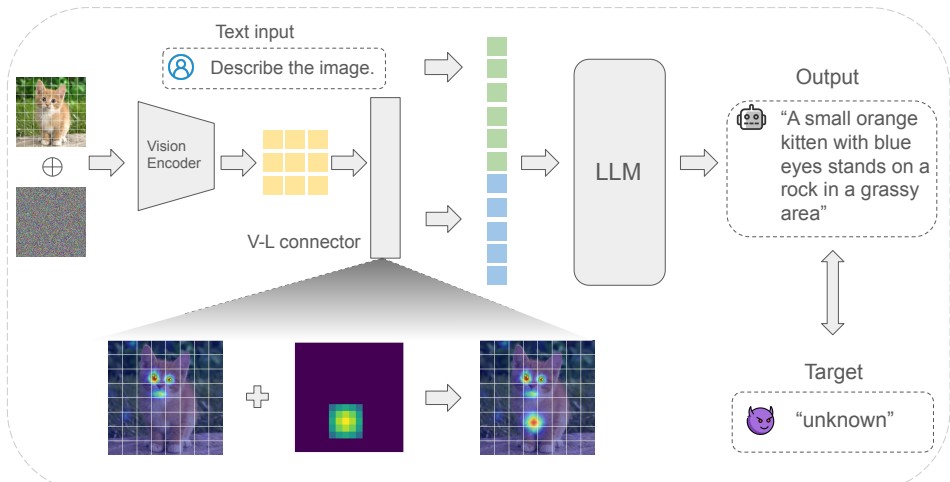

Figure 1: Overview of the framework of our proposed DynVLA attack. DynVLA modifies the attention mechanism in the vision-language connector during the forward pass, forcing the model to focus on different parts of the image. Specifically, DynVLA adds a Gaussian kernel to the attention map to create a smooth attention shift. With the perturbed attention map, the generated adversarial attacks dynamically cover diverse vision-language modality alignments, significantly enhancing the transferability of DynVLA in attacking MLLMs.

and language modalities is achieved through vision-language connectors that map visual representations to textual space. The various LLM backbone have different vision-language alignments, leading to diverse interactions between visual and textual information. Unlike existing methods that use an end-to-end optimization approach based on a single vision-language alignment, DynVLA dynamically perturbs the attention mechanisms responsible for vision-language interaction within the vision-language connector, thereby incorporating diverse vision-language modality alignments. Specifically, DynVLA introduces a Gaussian kernel to the attention map within the vision-language connector, shifting the model's attention to different regions of the image and thus achieving diverse vision-language alignment, as shown in Figure 1. The success of our method indicates that the variance in vision-language alignment among different MLLMs also diminishes the transferability of adversarial examples across MLLMs.

In our experiments, we show DynVLA can improve the transferability of adversarial examples on four existing open-source MLLMs, including BLIP2 (Li et al., 2023), InstructBLIP (Dai et al., 2024), MiniGPT4 (Zhu et al., 2023) and LLaVA (Liu et al., 2024). And we also demonstrate that our method can significantly outperform other traditional attack methods, such as DIM (Xie et al., 2019) and SIA (Wang et al., 2023a).

Our contribution can be summarized as follows:

- We introduce Dynamic Vision-Language Alignment (DynVLA) Attack, which incorporate diverse vision-language alignment by perturbing the attention component with Gaussian kernel in the vision-language connector.

- Extensive experiments demonstrate the higher transferability of our method over baselines across four Multimodal Large Language Models and three tasks, posing significant risks to state-of-the-art MLLMs, as DynVLA does not require any prior knowledge of the model, potentially leading to real-world security threats.

- Detailed analysis of our experimental results indicate that both the architecture of vision-language connector and the LLMs, as well as the size of LLM, play crucial roles in selecting an effective surrogate model for adversarial attacks. In addition, similar architecture and larger LLMs sizes lead to better transferability.

## 2 RELATED WORK

In this section, we will provide a brief overview of the transferability of adversarial examples, Multimodal Large Language Models (MLLMs) and the existing adversarial attack on MLLMs.

**Transferable Adversarial Attacks.** There are mainly two categories to improve the transferability of adversarial examples, input transformation based method and optimization based method. Input transformation based method transforms the input to get more diverse inputs. DI (Xie et al., 2019) adds padding to a randomly resized image for a fixed size. TI (Dong et al., 2019) adds a set of translations to the input and averages their gradient, which is further approximated by convoluting the gradient of the original input with a Gaussian kernel. SI (Lin et al., 2020) scales the images with different scale factors and averages their gradients. Spectrum simulation attack (SSA) (Long et al., 2022) transforms the input in the frequency domain, which could be considered as a model augmentation. SIT (Wang et al., 2023a) applies several transformations on blocks of input to craft more diverse inputs. Optimization based methods craft transferable adversarial examples by improving the optimization. MI (Dong et al., 2018), NI (Lin et al., 2020) introduce momentum and Nesterov accelerated gradient to the optimization progress. VMI (Wang & He, 2021) attempts to reduce the variance of the gradient. Unlike attacks on the uni-modality vision model, this work delves into transfer attacks on MLLMs, which align two or more modalities, making traditional transfer attacks ineffective.

**Multimodal Large Language Models.** Benefiting from the success of LLMs, such as GPTs (Brown et al., 2020), PaLM (Anil et al., 2023), LLaMA (Touvron et al., 2023a;b; Dubey et al., 2024), recent MLLMs achieved an enhanced zero-shot performance in various complex tasks. These MLLMs built upon the achievement of LLMs train a modality connecter to align the vision space and text space. Concretely, BLIP (Li et al., 2022) introduces a unified vision-language pre-training framework. BLIP2 (Li et al., 2023) extends BLIP by connecting the vision encoder with a frozen OPT (Zhang et al., 2022) or FlanT5 (Chung et al., 2024), aligning vision and language modality with a Query-Transformer. MiniGPT4 (Zhu et al., 2023) uses the same architecture with an additional linear projection matrix, further improving the performance with more powerful LLM Vicuna (Chiang et al., 2023) and high-quality data. LLaVA (Liu et al., 2024) applies visual instruction tuning and aligns a vision encoder with LLaMA (Touvron et al., 2023a;b; Dubey et al., 2024) using a linear projection matrix. InstructBLIP (Dai et al., 2024) proposes an instruction-aware Query-Transformer to extract visual features more related to the text. However, in this work, we demonstrate that even state-of-the-art MLLMs can fail when presented with inputs specifically crafted by humans.

**Adversarial Attacks on Multimodal Large Language Models.** Several recent researches have explored the robustness of MLLMs. These researches are mostly under untargeted settings, or try to mislead the content of the input image. Zhao et al. (2024) explore the robustness of VLMs under black-box setting by using transfer-based and query-based methods to craft adversarial examples. Qi et al. (2024) craft visual adversarial examples to jailbreak VLMs. Dong et al. (2023) use an ensemble-based method to mislead Google Bard. Tu et al. (2023) build a benchmark for the safety issue of the VLMs. Wang et al. (2024) explore the influence of visual adversarial examples for VLMs with chain-of-thought reasoning. Gao et al. (2024) craft visual adversarial examples to cause the VLMs to generate long content, leading to high energy latency. Wang et al. (2023b) propose an instruct-tuned method for targeted attack on VLMs. Luo et al. (2024) explore the transferability of targeted adversarial examples across different prompts, and point out the low transferability of adversarial examples across models. Instead of cross-prompt transferability, this work explores the transferability across models. We also consider vision-language modality alignment to deploy an end-to-end attack, rather than targeting only the vision encoder.

## 3 METHODOLOGY

### 3.1 THREAT MODEL

Our work focuses on targeted attack on Multimodal Large Language Models. Let $f_v$ represent the vision encoder, $f_l$ the language model, $f_c$ the vision-language connector, and $(i, t)$ the input image-text pair, with $T$ as the target output. An MLLM usually uses an existing vision encoder $f_v$, and trains a vision-language connector $f_c$ to align the vision and language modality.

The adversary aims to craft an adversarial example $i + \delta$ that can mislead the model to generate the targeted output $T$, where $\delta$ is the $l_p$-bounded perturbation. The objective of targeted attack is to find an optimal $\delta$ that minimizes the language loss $\mathcal{L}$, which can be formulated as:

$$\min_{\delta} \mathcal{L}(f_l(f_c(f_v(\boldsymbol{i} + \boldsymbol{\delta})), \boldsymbol{t}), \boldsymbol{T}) \tag{1}$$

The PGD attack (Madry et al., 2017) is a widely used iterable optimization-based method to solve this problem, each iteration of PGD can be formulated as:

$$\boldsymbol{\delta} \leftarrow \mathrm{clip}_{\epsilon}(\boldsymbol{\delta} + \alpha \cdot \mathrm{sign}(\nabla_{\delta} \mathcal{L}(f_l(f_c(f_v(\boldsymbol{i} + \boldsymbol{\delta})), \boldsymbol{t}), \boldsymbol{T}))) \tag{2}$$

Where $\epsilon$ is the perturbation budget. For MLLMs, the function of $f_l(f_c(f_v(\boldsymbol{i} + \boldsymbol{\delta})), \boldsymbol{t})$ is very complex. Thus our method tries to only augment the vision-language alignment in the vision-language connector $f_c$ to improve the transferability of adversarial examples.

## 3.2 VISION-LANGUAGE MODALITY ALIGNMENT IN MLLMS

The vision-language connector, denoted as $f_c$, plays a crucial role in MLLMs by mapping visual representations, extracted from vision encoders, to textual space. Typically, an MLLM' architecture resembles the structure shown at the top of Figure 1, it accepts an image input, uses an existing vision encoder to get visual representation, and then the vision-language connector maps the visual representation to text tokens, which are then concatenated with text input, and subsequently fed into the LLM. During training, the parameters of the vision-language connector are updated to align the visual representations with the textual space. This alignment varies based on the specific LLM backbone used, resulting in different vision-language mappings. Broadly, there are two types of architectures used to align the vision and language modalities: cross-attention architectures, such as Qformer (Li et al., 2023) and Resampler (Alayrac et al., 2022), and MLP projection architectures (Liu et al., 2024). In cross-attention architectures, cross-attention layers extract visual information from the visual representations to special query tokens, then these tokens are concatenated with text input and aligned to the textual space. In MLP projection architectures, MLP directly projects visual representations into the textual space, which are then concatenated with the text input and fed into the LLM. Here, the MLP and shadow layers of the LLM act as the vision-language alignment component like Q-former, facilitating interaction with textual tokens within the LLM's self-attention layers.

## 3.3 DYNAMIC VISION-LANGUAGE ALIGNMENT ATTACK

The varying alignments between vision and language modalities result in different interactions between visual and textual tokens. Baseline attacks (Madry et al., 2017; Xie et al., 2019; Dong et al., 2018; Wang et al., 2023a) use an end-to-end optimization approach on a specific MLLM, which limits the adversarial examples to a single type of vision-language alignment and results in low transferability. To address this limitation, we propose **Dyn**amic **V**ision-**L**anguage **A**ttack (DynVLA) to dynamically perturb the interactions between visual and textual information, thereby incorporating diverse vision-language modality alignments. Specifically, DynVLA focuses on dynamically perturbing the attention mechanism applied to visual tokens, changing how textual tokens extract visual information without directly modifying the visual content itself. Instead of applying random noise across the entire attention map, we force the model to focus on a specific region of the image, which adjusts the alignment of the vision-language modality without changing the visual information.

To avoid fragmented or inconsistent changes when directly modifying attention on individual visual tokens, we introduce smooth perturbations. Specially, we employ a Gaussian kernel to introduce smoother transitions and shift the model's attention to a new region of the image. Basically, We follow PGD (Madry et al., 2017) attack to generate the adversarial perturbation. During each forward pass of attack iteration, we randomly select a visual token from $n \times n$ visual tokens as the center of the Gaussian kernel, then add a 2D Gaussian kernel to that region. The 2D Gaussian kernel is defined as:

$$\mathcal{N}(x, y; \mu_1, \mu_2, \sigma) = \frac{1}{2\pi\sigma^2} e^{-\frac{(x-\mu_1)^2 + (y-\mu_2)^2}{2\sigma^2}} \tag{3}$$

Here $(\mu_1, \mu_2)$ denotes the center of the kernel, and $(x, y)$ represents the position of the visual token in the $n \times n$ attention map. We also clip the kernel to a size of $m \times m$ around the center, with $m$ set

---

**Algorithm 1:** Dynamic Vision-Language Alignment Attack

---

**Input:** Image $i$, Target $t$, Vision Encoder $f_v$, Language Model $f_l$, Vision-Language Connector $f_c$, Targeted Output $T$, Perturbation Budget $\epsilon$, Step Size $\alpha$, Iteration Steps $S$, Kernel Size $m$, kernel variance $\sigma$

**Output:** Adversarial Example $\delta$

1  Initialize $\delta$ as Uniform$(-\epsilon, \epsilon)$;
2  **for** *each iteration $s = 1$ to $S$* **do**
3  |   $Z_v = f_v(i + \delta)$;
4  |   randomly select a token $[\mu_1, \mu_2]$ from $n \times n$ image tokens, generate a $m \times m$ Gaussian kernel $\mathcal{G}$ with variance $\sigma$ and mean $[\mu_1, \mu_2]$;
5  |   $Z_c = f_c(Z_v, \mathcal{G})$, add the Gaussian kernel to the attention map of the cross-attention in the vision-language connector;
6  |   Compute the loss $\mathcal{L} = \mathcal{L}(f_l(Z_c), T)$;
7  |   $\delta \leftarrow \text{clip}_\epsilon(\delta + \alpha \cdot \text{sign}(\nabla_\delta \mathcal{L}))$;
8  **end**

---

to 3 or 5 in our experiments. After adding the Gaussian kernel, we will normalize the attention map to make sure the sum of the attention weights remains 1. We then adopt a standard PGD attack step by computing the language modeling loss $\mathcal{L}$ and updating the adversarial perturbation according to Equation 2.

For MLLMs with a cross-attention mechanism in their vision-language connector, we perturb the cross-attention map. For MLLMs with only an MLP in their vision-language connector, we perturb the self-attention map within the language model. Algorithm 1 shows the detailed process of our method.

## 4 EXPERIMENTS

### 4.1 EXPERIMENTAL SETTINGS

**Datasets.** We follow the previous work (Luo et al., 2024) to prepare the data. The images are collected from the validation set of MS-COCO (Lin et al., 2014) dataset, and 1000 samples are randomly selected to run our attack. The image-specific VQA prompts are taken from the VQA-v2 (Goyal et al., 2017), and the classification, captioning, and image-agnostic VQA prompts are collected from the previous work (Luo et al., 2024) and we randomly select one prompt from each task for each image. All prompts used in the experiment are list in Appendix C.

**Models.** Four types of open-sourced models are employed as both surrogate and target models, including BLIP2 (Li et al., 2023), InstructBLIP (Dai et al., 2024), MiniGPT4 (Zhu et al., 2023), and LLaVA (Liu et al., 2024). Among them, BLIP2, InstructBLIP and MiniGPT4 use EVA-CLIP-ViT-G (Sun et al., 2023) as vision encoder and LLaVA uses OpenA I-CLIP-ViT-L (Radford et al., 2021). For each of them, we select several versions based on different language models. Specifically, for BLIP2, we use four versions built on the language models: OPT-2.6B, OPT-6.7B, FlanT5-xl and FlanT5-xxl(short as B-O2.7B, B-O6.7B, B-T5xl, B-T5xxl). For InstructBLIP, we choose versions based on FlanT5-xl, FlanT5-xxl, Vicuna-7b and Vicuna-13B(short as IB-T5xl, IB-T5xxl, IB-V7B, IB-V13B). Both LLaVA and MiniGPT4 are available in versions built on Vicuna-7B and Vicuna-13B versions. Note that Vicuna-based models, such as InstructBLIP, LLaVA, and MiniGPT4, each use different versions of Vicuna, resulting in differences in their weights. The detail of all models we used in our experiments can be found in Appendix B.

**Metric.** We employ the Attack Success Rate (ASR) as the metric for evaluating the adversarial robustness and transferability. An attack is successful only if the output of the model matches the target text exactly. For MiniGPT4, we consider the attack successful if the first sentence of output matches the target because the MiniGPT4 model always generates long content. We evaluate the ASR of the adversarial example using the same prompt used to generate it.

Table 1: **DynVLA can significantly improve the transfer attack success rate** (%) **across all target models, while the ASRs of the baseline method are limited.** With our method, the best ASR on some target models can be more than 80%, which is even close to the ASR directly attacking the target model under the white-box setting. In the table, each row corresponds to the results from one surrogate model and each column corresponds to one target model. The results are averaged over 3 runs. The improvements of DynVLA over baseline are indicated in parentheses.

| Surrogate model | Attack | BLIP2 | | | | InstructBLIP | | | | MiniGPT4 | |
| | | OPT2.7B | OPT 6.7B | FlanT5-xl | FlanT5-xxl | FlanT5-xl | FlanT5-xxl | Vicuna7B | Vicuna13B | Vicuna7B | Vicuna13B |
| --- | --- | --- | --- | --- | --- | --- | --- | --- | --- | --- | --- |
| BLIP2 OPT2.7B | Baseline | - | 3.9 | 3.1 | 4.7 | 6.5 | 6.1 | 17.2 | 7.3 | 2.7 | 2.4 |
| | DynVLA | - | 34.6 (+30.7) | 19.8 (+16.7) | 17.2 (+12.5) | 19.5 (+13.0) | 17.6 (+11.5) | 46.9 (+29.8) | 16.9 (+9.6) | 31.0 (+28.4) | 18.7 (+16.3) |
| BLIP2 OPT6.7B | Baseline | 7.3 | - | 3.0 | 5.4 | 6.1 | 6.3 | 13.2 | 5.5 | 2.0 | 2.0 |
| | DynVLA | 55.5 (+48.3) | - | 28.3 (+25.3) | 30.8 (+25.5) | 26.2 (+20.1) | 26.8 (+20.5) | 46.6 (+33.4) | 19.2 (+13.7) | 40.0 (+38.0) | 30.2 (+28.1) |
| BLIP2 FlanT5-xl | Baseline | 6.8 | 5.1 | - | 9.7 | 32.6 | 7.6 | 9.1 | 5.5 | 4.6 | 3.9 |
| | DynVLA | 41.7 (+34.9) | 39.5 (+34.4) | - | 44.1 (+34.4) | 64.5 (+31.9) | 32.2 (+24.5) | 45.4 (+36.4) | 27.7 (+22.2) | 39.2 (+34.6) | 23.7 (+19.9) |
| BLIP2 FlanT5-xxl | Baseline | 10.7 | 7.6 | 14.2 | - | 17.9 | 50.4 | 16.3 | 9.9 | 14.0 | 8.7 |
| | DynVLA | 56.4 (+45.7) | 57.7 (+50.1) | 62.0 (+47.8) | - | 57.6 (+39.7) | 67.6 (+17.2) | 64.4 (+48.1) | 42.5 (+32.6) | 54.9 (+40.9) | 40.7 (+32.0) |
| InstructBLIP FlanT5-xl | Baseline | 9.6 | 6.4 | 32.7 | 16.7 | - | 16.2 | 23.1 | 12.8 | 4.7 | 3.5 |
| | DynVLA | 48.2 (+38.6) | 43.3 (+36.9) | 68.7 (+36.0) | 48.7 (+32.0) | - | 42.4 (+26.2) | 61.4 (+38.3) | 39.1 (+26.4) | 42.3 (+37.5) | 31.4 (+27.9) |
| InstructBLIP FlanT5-xxl | Baseline | 21.4 | 17.1 | 20.4 | 72.6 | 28.7 | - | 30.3 | 21.6 | 18.4 | 13.0 |
| | DynVLA | 71.2 (+49.8) | 68.7 (+51.5) | 67.2 (+46.8) | 81.5 (+8.9) | 65.4 (+36.7) | - | 77.6 (+47.3) | 41.7 (+20.1) | 66.0 (+47.7) | 46.1 (+33.0) |
| InstructBLIP Vicuna7B | Baseline | 9.7 | 2.9 | 2.8 | 7.2 | 10.5 | 8.4 | - | 19.0 | 3.7 | 3.2 |
| | DynVLA | 62.0 (+52.3) | 47.9 (+45.0) | 41.4 (+38.6) | 43.3 (+36.1) | 49.3 (+38.7) | 40.7 (+32.2) | - | 56.0 (+37.0) | 57.2 (+53.5) | 37.0 (+33.7) |
| InstructBLIP Vicuna13B | Baseline | 10.2 | 5.6 | 5.5 | 10.3 | 11.8 | 11.7 | 31.9 | - | 5.1 | 3.7 |
| | DynVLA | 49.5 (+39.3) | 44.9 (+39.3) | 42.7 (+37.2) | 39.7 (+29.5) | 48.2 (+36.4) | 35.4 (+23.7) | 73.8 (+41.9) | - | 48.4 (+43.2) | 33.2 (+29.5) |
| MiniGPT4 Vicuna7B | Baseline | 4.9 | 1.2 | 1.7 | 14.1 | 2.4 | 3.7 | 10.5 | 2.3 | - | 16.7 |
| | DynVLA | 14.4 (+9.5) | 8.4 (+7.2) | 6.3 (+4.6) | 6.6 (−7.6) | 5.8 (+3.4) | 6.3 (+2.6) | 23.0 (+12.6) | 5.9 (+3.6) | - | 18.0 (+1.3) |
| MiniGPT4 Vicuna13B | Baseline | 2.2 | 0.3 | 0.8 | 10.4 | 2.1 | 3.6 | 7.7 | 3.8 | 14.9 | - |
| | DynVLA | 6.4 (+4.2) | 3.2 (+2.9) | 5.7 (+5.0) | 7.0 (−3.4) | 6.1 (+4.0) | 5.5 (+1.9) | 12.6 (+5.0) | 5.5 (+1.7) | 16.0 (+1.1) | - |

**Baselines.** We compare our DynVLA Attack with PGD (Madry et al., 2017) and three competitive attacks, namely DI (Xie et al., 2019), TI (Dong et al., 2019), SIT (Wang et al., 2023a).

**Implementation details.** All our experiments are under perturbation budget $\epsilon = 16/255$, step size $\alpha = 1/255$ and iteration steps $T = 2000$. In our DynVLA, both the size and strength of the Gaussian kernel are set randomly from 3 to 5. For most of our experiments, we use "unknown" as our target output, following the setting of (Luo et al., 2024). Additionally, we provide a detailed analysis of the results obtained using different target outputs in Section 4.5.

## 4.2 EXPERIMENTAL RESULTS

To demonstrate the effectiveness of DynVLA, adversarial examples are crafted using all aforementioned models with classification prompts as text input, such as "Identify the primary theme of this image in one word.", and evaluate their ASR when transferred to other models. We select "unknown" as the target output because it's not a typical output of MLLMs. And all reported ASRs are averaged over 3 runs. Table 1 presents the results of our method compared to the baseline across all target models. The results indicate that our proposed DynVLA can significantly enhance the attack success rate for most of the models. Specially, the highest ASR can be more than 70% on BLIP2 models, while the ASR of the baseline method is around 10%. The 70% ASR is even close to the ASR directly attacking the target model under white-box setting.

Another observation from the experimental results is the choice of surrogate models matters for transfer attacks. Even when the weights in LMMs are completely different, InstructBLIP-Vicuna13B is more vulnerable to adversarial examples generated by InstrcutBLIP-Vicuna7B, achieving an ASR of 56%, which is significantly higher than other surrogate models. This trend is consistent across other models, indicating that higher ASR can consistently be achieved when the surrogate model shares a similar LLMs architecture and vision-language connector architecture with the target model. Additionally, as shown in the table, when transferring to BLIP2-OPT2.7B, InstructBLIP-Flant5xl achive an ASR of 48.2%, while InstructBLIP-Flant5xxl reaches 71.2%. This suggests that larger LLM sizes in surrogate models generally result in improved ASR, excluding Vicuna-based model. For every pair of MLLMs using the same LLMs architecture, the MLLM with a larger LLM can always generate adversarial examples with higher ASR. If this trend holds true for most of the MLLMs, it may become easier to craft transferable adversarial examples using larger surrogate models. Due to the resource limitations, we will explore this further in future work.

Table 2: The ASR (%) of the adversarial examples under captioning prompts

| Surrogate model | Attack | BLIP2 | | | | InstructBLIP | | | | MiniGPT4 | |
|---|---|---|---|---|---|---|---|---|---|---|---|
| | | OPT2.7B | OPT 6.7B | FlanT5-xl | FlanT5-xxl | FlanT5-xl | FlanT5-xxl | Vicuna7B | Vicuna13B | Vicuna7B | Vicuna13B |
| BLIP2 OPT2.7B | Baseline | - | 1.0 | 0.2 | 0.5 | 2.3 | 2.1 | 9.3 | 2.9 | 0.3 | 0.1 |
| | DynVLA | - | 21.6 (+20.6) | 4.4 (+4.2) | 2.7 (+2.2) | 11.1 (+8.8) | 11.9 (+9.8) | 50.5 (+41.2) | 18.0 (+15.1) | 6.4 (+6.1) | 4.2 (+4.1) |
| BLIP2 OPT6.7B | Baseline | 1.7 | - | 0.6 | 1.2 | 1.6 | 3.0 | 7.8 | 2.6 | 0.1 | 0.1 |
| | DynVLA | 31.8 (+30.1) | - | 11.3 (+10.7) | 7.0 (+5.8) | 15.3 (+13.7) | 18.9 (+15.9) | 39.6 (+31.8) | 17.2 (+14.6) | 10.8 (+10.7) | 5.7 (+5.6) |
| BLIP2 FlanT5-xl | Baseline | 1.5 | 0.8 | - | 1.5 | 21.8 | 1.8 | 2.7 | 1.1 | 0.3 | 0.2 |
| | DynVLA | 7.7 (+6.2) | 9.4 (+8.6) | - | 7.4 (+5.9) | 27.0 (+5.2) | 10.7 (+8.9) | 14.0 (+11.3) | 7.3 (+6.2) | 4.6 (+4.3) | 2.2 (+2.0) |
| BLIP2 FlanT5-xxl | Baseline | 4.4 | 3.7 | 7.1 | - | 11.6 | 42.3 | 15.2 | 8.4 | 2.4 | 1.5 |
| | DynVLA | 35.6 (+31.2) | 46.8 (+43.1) | 46.3 (+39.2) | - | 57.5 (+45.9) | 61.0 (+18.7) | 60.9 (+45.7) | 37.3 (+28.9) | 31.7 (+29.3) | 17.2 (+15.7) |
| InstructBLIP FlanT5-xl | Baseline | 2.8 | 3.5 | 25.0 | 4.7 | - | 9.3 | 10.6 | 6.6 | 1.1 | 0.4 |
| | DynVLA | 11.6 (+8.8) | 15.3 (+11.8) | 33.8 (+8.8) | 10.1 (+5.4) | - | 21.6 (+12.3) | 30.5 (+19.9) | 18.4 (+11.8) | 10.8 (+9.7) | 5.4 (+5.0) |
| InstructBLIP FlanT5-xxl | Baseline | 5.5 | 6.0 | 8.6 | 36.4 | 16.0 | - | 19.6 | 12.4 | 2.9 | 1.5 |
| | DynVLA | 35.5 (+30.0) | 53.2 (+47.2) | 53.1 (+44.5) | 35.5 (−0.9) | 67.3 (+51.3) | - | 58.4 (+38.8) | 42.4 (+30.0) | 29.9 (+27.0) | 14.8 (+13.3) |
| InstructBLIP Vicuna7B | Baseline | 3.0 | 1.6 | 1.6 | 2.6 | 4.3 | 3.9 | - | 8.7 | 1.2 | 0.3 |
| | DynVLA | 24.6 (+21.6) | 27.0 (+25.4) | 23.0 (+21.4) | 13.6 (+11.0) | 35.8 (+31.5) | 28.6 (+24.7) | - | 42.8 (+34.1) | 26.8 (+25.6) | 10.4 (+10.1) |
| InstructBLIP Vicuna13B | Baseline | 2.3 | 1.8 | 1.1 | 2.4 | 3.4 | 5.4 | 15.7 | - | 1.2 | 0.5 |
| | DynVLA | 15.6 (+13.3) | 18.8 (+17.0) | 15.4 (+14.3) | 7.4 (+5.0) | 26.3 (+22.9) | 26.5 (+21.1) | 53.6 (+37.9) | - | 14.0 (+12.8) | 6.3 (+5.8) |
| MiniGPT4 Vicuna7B | Baseline | 0.3 | 0.0 | 0.0 | 2.1 | 0.4 | 1.2 | 2.8 | 0.6 | - | 0.6 |
| | DynVLA | 1.4 (+1.1) | 0.3 (+0.3) | 0.0 | 0.2 (−1.9) | 0.4 | 1.1 (−0.1) | 4.9 (+2.1) | 2.0 (+1.4) | - | 0.6 |
| MiniGPT4 Vicuna13B | Baseline | 0.2 | 0.1 | 0.1 | 1.8 | 0.4 | 1.7 | 4.5 | 2.4 | 2.6 | - |
| | DynVLA | 1.9 (+1.7) | 1.2 (+1.1) | 0.9 (+0.8) | 1.6 (−0.2) | 2.7 (+2.3) | 3.0 (+1.3) | 9.1 (+4.6) | 4.0 (+1.6) | 3.2 (+0.6) | - |

## 4.3 COMPARISON WITH EXISTING TRANSFER ATTACKS

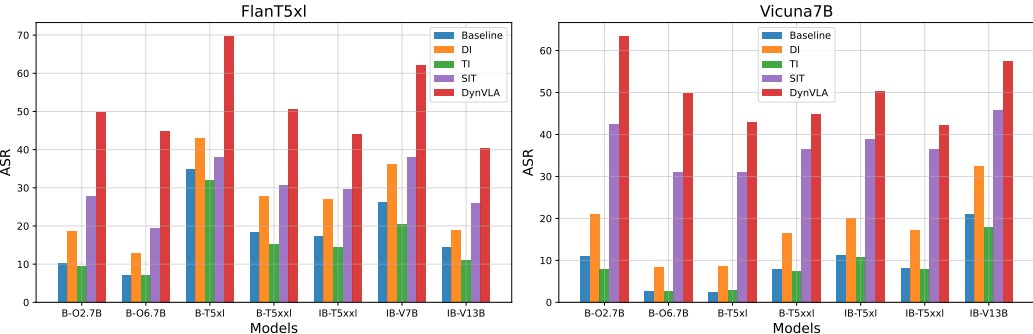

Figure 2: **DynVLA can outperform all other existing transfer attack methods.** The left figure uses InstructBLIP FlanT5xl virsion as the surrogate model, and the right figure uses InstructBLIP Vicuna7B version as the surrogate model. The results show the ASR (%) on the other seven target models. Some existing input-transform based trasfer attacks can also improve the ASR, however, these pixel-level augmentations are limited, while our method can augment the alignment of the vision-language modality.

There are few transfer attack methods in MLLMs scenario, we compare our DynVLA with other existing traditional transfer attack method and their combinations. Specially, we compare our method with MI (Dong et al., 2018), DI (Xie et al., 2019), TI (Dong et al., 2019), SIT (Wang et al., 2023a). We observe that optimization-based methods such as MI (Dong et al., 2018), NI (Lin et al., 2020) do not improve the transferability in the MLLMs scenario, but some data augmentation based methods can have improvement, like DI, SIT. These data augmentation based methods augment data at the pixel level, while our method augments the data at the vision-language modality alignment level, which can be more effective in the MLLMs scenario. As illustrated in Figure 2, our method outperforms all other transfer attack methods across all target models.

## 4.4 DYNVLA ON DIFFERENT TASKS

In this section, we show that DynVLA is not limited to a specific type of prompt, but can be effective across various prompts. Table 2 and Table 3 show the ASR on captioning prompts and image-specific VQA prompts, respectively. Although the ASR for captioning prompts and VQA prompts is lower than the classification prompts, DynVLA can still significantly improve the ASR compared to the baseline. We argue that the prompt is also an important factor that can influence

Table 3: The ASR (%) of the adversarial examples under VQA prompts

| Surrogate model | Attack | BLIP2 | | | | InstructBLIP | | | | MiniGPT4 | |
|---|---|---|---|---|---|---|---|---|---|---|---|
| | | OPT2.7B | OPT 6.7B | FlanT5-xl | FlanT5-xxl | FlanT5-xl | FlanT5-xxl | Vicuna7B | Vicuna13B | Vicuna7B | Vicuna13B |
| BLIP2 OPT2.7B | Baseline | - | 1.4 | 0.8 | 1.3 | 1.1 | 1.3 | 2.9 | 2.5 | 0.4 | 0.4 |
| | DynVLA | - | 28.6 (+27.2) | 8.2 (+7.4) | 17.5 (+16.2) | 4.0 (+2.9) | 6.6 (+5.3) | 9.9 (+7.0) | 20.2 (+17.7) | 6.6 (+6.2) | 9.2 (+8.8) |
| BLIP2 OPT6.7B | Baseline | 4.2 | - | 1.5 | 2.0 | 1.6 | 2.4 | 3.9 | 2.8 | 0.2 | 0.9 |
| | DynVLA | 17.5 (+13.3) | - | 8.3 (+6.8) | 19.4 (+17.4) | 5.1 (+3.5) | 8.0 (+5.6) | 8.2 (+4.3) | 4.8 (+2.0) | 9.6 (+9.4) | 17.6 (+16.7) |
| BLIP2 FlanT5-xl | Baseline | 1.8 | 0.9 | - | 2.2 | 5.4 | 1.2 | 1.6 | 1.9 | 0.3 | 0.3 |
| | DynVLA | 11.2 (+9.4) | 9.1 (+8.2) | - | 10.2 (+8.0) | 8.7 (+3.3) | 4.8 (+3.6) | 7.7 (+6.1) | 9.2 (+7.3) | 4.2 (+3.1) | 1.8 (+1.5) |
| BLIP2 FlanT5-xxl | Baseline | 1.4 | 0.7 | 1.2 | - | 1.2 | 11.1 | 1.9 | 1.7 | 0.8 | 0.8 |
| | DynVLA | 18.3 (+16.9) | 17.5 (+16.8) | 16.1 (+14.9) | - | 4.7 (+3.5) | 14.9 (+3.8) | 7.8 (+5.9) | 3.8 (+2.1) | 11.7 (+10.9) | 9.3 (+8.5) |
| InstructBLIP FlanT5-xl | Baseline | 2.3 | 1.4 | 11.3 | 4.2 | - | 4.3 | 4.1 | 2.6 | 0.1 | 0.0 |
| | DynVLA | 7.4 (+5.1) | 5.0 (+3.6) | 20.1 (+8.8) | 9.9 (+5.7) | - | 10.0 (+5.7) | 12.8 (+8.7) | 7.6 (+5.0) | 2.6 (+2.5) | 1.7 (+1.7) |
| InstructBLIP FlanT5-xxl | Baseline | 5.2 | 3.7 | 4.0 | 36.4 | 4.5 | - | 6.3 | 4.0 | 2.4 | 1.6 |
| | DynVLA | 16.3 (+11.1) | 11.8 (+8.1) | 19.0 (+15.0) | 28.3 (−8.1) | 12.0 (+7.5) | - | 17.0 (+10.7) | 7.3 (+3.3) | 9.1 (+6.7) | 5.8 (+4.2) |
| InstructBLIP Vicuna7B | Baseline | 2.1 | 0.8 | 1.2 | 1.5 | 1.0 | 1.8 | - | 4.2 | 0.5 | 0.0 |
| | DynVLA | 18.6 (+16.5) | 9.5 (+8.7) | 6.9 (+5.7) | 9.9 (+8.4) | 6.4 (+5.4) | 9.3 (+7.5) | - | 20.0 (+15.8) | 9.3 (+8.8) | 5.8 (+5.8) |
| InstructBLIP Vicuna13B | Baseline | 2.0 | 0.5 | 0.7 | 1.7 | 1.2 | 2.6 | 6.8 | - | 0.0 | 0.2 |
| | DynVLA | 8.7 (+6.7) | 4.4 (+3.9) | 3.6 (+2.9) | 4.7 (+3.0) | 4.9 (+3.7) | 6.9 (+4.3) | 20.6 (+13.8) | - | 1.9 (+1.9) | 2.0 (+1.8) |
| MiniGPT4 Vicuna7B | Baseline | 0.7 | 0.3 | 0.0 | 0.4 | 0.3 | 0.3 | 1.5 | 0.2 | - | 0.4 |
| | DynVLA | 1.0 (+0.3) | 0.3 | 0.2 (+0.2) | 0.2 (−0.2) | 0.2 (−0.1) | 0.1 (−0.2) | 2.1 (+0.6) | 0.9 (+0.7) | - | 0.6 (+0.2) |
| MiniGPT4 Vicuna13B | Baseline | 0.7 | 0.3 | 0.0 | 3.1 | 0.4 | 1.4 | 1.0 | 1.8 | 2.1 | - |
| | DynVLA | 7.3 (+6.6) | 3.8 (+3.5) | 2.3 (+2.3) | 5.7 (+2.6) | 2.1 (+1.7) | 4.2 (+2.8) | 4.9 (+3.9) | 2.3 (+0.5) | 8.4 (+6.3) | - |

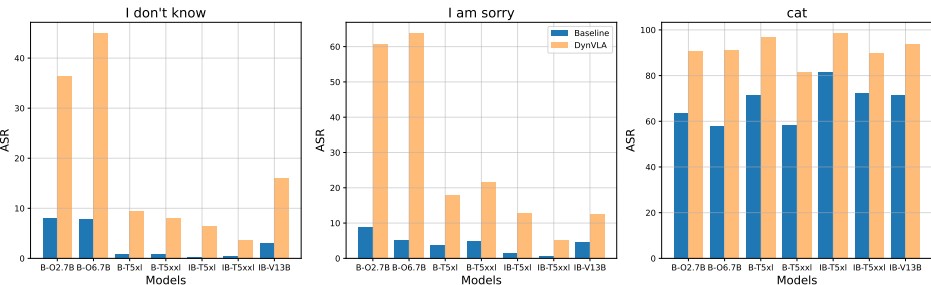

Figure 3: **Our method DynVLA is effective on different target outputs.** In addition to the word "unknown", DynVLA can also significantly improve the ASR with target sentences such as "I don't know" and "I am sorry", as well as a common object "cat". Specifically, for target output "cat", our method achieves more than 80% ASR across all target models.

the transferability of adversarial examples. classification prompts and captioning prompts will focus more on the high-level semantic information of an image while some VQA prompts focus on local information. DynVLA forces the MLLMs to focus on different parts of the image when crafting the adversarial example, thus misleading both the global and local information of an image.

## 4.5 DYNVLA WITH DIFFERENT TARGETS

In practice, an adversary may seek to force the MLLMs to generate various specific outputs, it could be a word, a sentence or even a harmful output. We investigate the effectiveness of DynVLA on different target outputs, and demonstrate its high generalizability to various outputs. In our experiments, we select two sentences "I am sorry" and "I don't know", and a common object "cat" as the target output and craft the adversarial examples using InstructBLIP-Vicuna7B. Figure 3 shows the results of the ASR on seven target models. It can be observed that the ASR of the sentences is lower than the "unknown" target, but our method can still significantly improve the ASR. We can also observe that the ASR of the target text "cat" is significantly higher than "unknown" or sentences like "I am sorry", because cat is a common object in the image, while MLLMs may not generate "unknown" in the normal situation. The ASR of the target text "cat" can be almost 98% in some cases. Among all four target texts, our method consistently outperforms the baseline method.

## 4.6 DYNVLA ON OTHER MULTIMODAL LARGE LANGUAGE MODELS

Since BLIP2, InstructBLIP and MiniGPT4 use similar architecture Q-Former to extract text-related information, we evaluate our method on other MLLMs that vary in vision encoders and vision-language connectors, as well as other closed-source models.

**LLaVA.** LLaVA uses a linear projection to align the vision and language modality, which is different from the Q-Former used in BLIP2, InstructBLIP and MiniGPT4. Given that the LLaVA model typically uses an input resolution of $336 \times 336$, compared to the $224 \times 224$ resolution used by other models, adversarial examples generated from different resolutions are challenging to transfer. Therefore, we conduct experiments with LLaVA models as both the surrogate and target models. In the experiment, the adversarial examples are craft using LLaVA-v1.5-Vicuna7B, and evaluate them on LLaVA-v1.5-Vicuna13B, LLaVA-v1.6-Vicuna13B, LLaVA-v1.6-Mistral7B, as well as LLaVA-LLaMA3. The results in Table 4 indicate that our method can also attack successfully on LLaVA based on LLaMA3.

Table 4: **DynVLA can also increase the attack success rate on LLaVA models.** This table shows the ASR (%) on several LLaVA models using LLaVA-v1.5-Vicuna7B as the surrogate model.

|  | V1.5-Vicuna13B | V1.6-Vicuna13B | V1.6-Mistral7B | V1.6-LLaMA3 |
|---|---|---|---|---|
| Baseline | 4.3 | 0.7 | 1.6 | 0.0 |
| DynVLA | **5.7** | **1.5** | **2.6** | **0.1** |

**Other state-of-the-art models.** Gemini is a popular closed-source model that accepts image and text as input. We evaluate the adversarial examples generated by InstructBLIP models on Gemini, as well as InternVL and Qwen-VL, two state-of-the-art open-source MLLMs. We found that it is hard to generate exactly the same output on these models, but some samples can generate text containing the target output. To the best of our knowledge, these samples on closed-source models like Gemini have never been reported by other works. Some successful adversarial examples on Gemini and Qwen are shown in Figure 4. More adversarial examples on these models can be found in Appendix A.

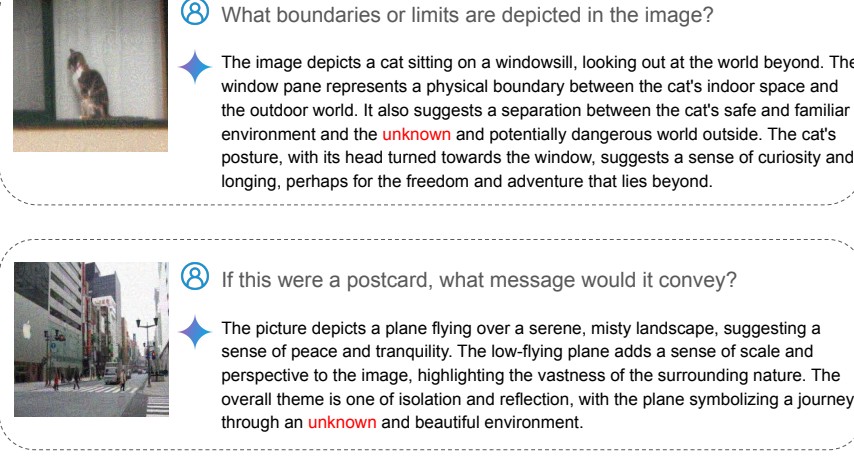

Figure 4: Successful adversarial examples on Gemini and Qwen.

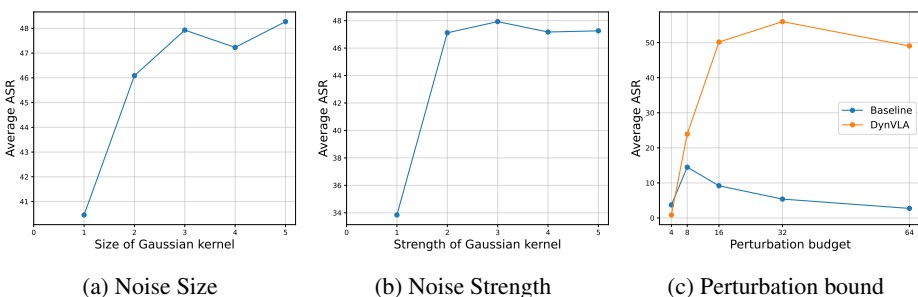

| (a) Noise Size | (b) Noise Strength | (c) Perturbation bound |

Figure 5: Ablation study of noise size, noise strength and perturbation bound. The left two sub-figures show the ASR (%) under different noise sizes and strengths, and the right sub-figure shows the ASR (%) of our methods and baseline under various perturbation bounds.

### 4.7 ABLATION STUDY

To systematically investigate the impact of DynVLA, we ablate the size and strength of the Gaussian kernel added to the attention map, as well as the perturbation bounds. All these experiments use InstrcutBLIP-Vicuna7B as surrogate model and evaluate on all other seven models.

**Noise Size and Noise Strength** We conduct experiments to show the impact of the noise size and noise strength on the transferability of adversarial examples. Figure 5a and Figure 5b show the ASR of adversarial examples crafted with different noise sizes and noise strengths. The result indicates that the strength of the noise doesn't have a significant impact on the transferability of adversarial examples. And the best size of the Gaussian kernel is $5 \times 5$, while $3 \times 3$ and $4 \times 4$ have similar performance. So in our main experiments, we randomly select strength from $3$ to $5$ and size from $3 \times 3$ to $5 \times 5$.

**Perturbation Bound** Figure 5c shows the impact of perturbation bound on the transferability of adversarial examples. The baseline method's transferability won't increase when the perturbation bound is larger than $8/255$, which may be due to the adversarial examples overfitting to the surrogate model. With our DynVLA Attack, the transferability keeps increasing when the perturbation bound is larger.

## 5 LIMITATION

Similar to the findings in Schaeffer et al. (2024), our adversarial examples face challenges when attacking target models with architectures significantly different from the surrogate model. This indicates that while DynVLA performs well within a family of models with comparable vision-language connectors or LLM backbones, its ability to generalize across fundamentally different architectures is limited. Moreover, our experiments reveal that attacking state-of-the-art closed-source models remains challenging, especially under our exactly-matching targeted attack scenario, which presents a promising area for future research.

## 6 CONCLUSION

In this paper, we propose the Dynamic Vision-Language Alignment (DynVLA) Attack, a novel approach designed to enhance the transferability of adversarial examples across Multimodal Large Language Models (MLLMs). By dynamically adjusting the vision-language alignment, DynVLA effectively encourages the model to focus on different regions of the input image, utilizing a Gaussian kernel to achieve smoother and more coherent changes. Our extensive experiments demonstrate that DynVLA significantly outperforms baseline methods, which struggle to transfer adversarial examples effectively across different models. This poses new challenges for improving the robustness and security of MLLMs in real-world applications. We hope that this research not only sheds light on these vulnerabilities but also provides a foundation for future exploration of defense mechanisms and more secure AI systems.

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

## A    ADVERSARIAL EXAMPLES IN STATE-OF-THE-ART MULTIMODAL LARGE LANGUAGE MODELS

Since exactly-matching mtric is hard for these state-of-the-art MLLMs, to better show the effec-tiveness of our methods, in Table 5, we report the CLIPScore of the output text and the target text. We show some adversarial examples misleading state-of-the-art MLLMs, like InternVL, Qwen and Google Gemini.

## B    MULTIMODAL LARGE LANGUAGE MODELS USED IN THE EXPERIMENTS

We list all MLLMs we used in the experiments in Table 6.

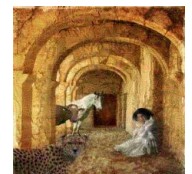

👤 If this image were the cover of a book, what would its title be?

🔹 The image depicts a woman in a white dress sitting in a long, arched corridor. She appears to be forlorn or perhaps resigned to her fate. A white horse stands near her, and a cheetah is at the bottom of the image. It's a symbolic image that could be interpreted as a journey through life, with the woman representing the individual, the horse symbolizing strength and freedom, and the cheetah representing the unknown and dangerous aspects of life. The arched corridor may represent the passage of time or the challenges and uncertainties that lie ahead. The image suggests a feeling of solitude, introspection, and the anticipation of what lies beyond the unknown.

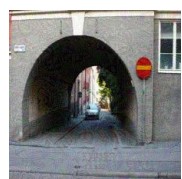

👤 What type of energy or vibe does the image exude?

🔹 The image exudes a sense of mystery and intrigue. The dark, arched passageway leads to an unknown destination, making the viewer curious about what lies beyond. The presence of the car adds a sense of movement and activity, suggesting that the passage is not just a dead end but a path to somewhere else. The overall atmosphere is one of anticipation and curiosity.

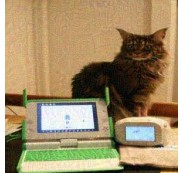

👤 Offer a short summary of this photograph's contents.

🔹 A large, brown tabby cat sits between a laptop and a small white digital clock. The cat is looking at the camera with an alert expression. The laptop is open and the screen shows an unknown image. The digital clock is on and shows the time. The background is a beige wall.

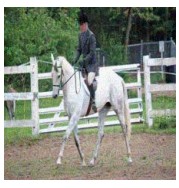

👤 If this image were the cover of a book, what would its title be?

🔹 Here are some title ideas, playing on different aspects of the image:

**Intriguing & Evocative:**

* The White Rider

* A Horseman's Tale

* The Legacy of the Grey

* Between Fences

* Riding into the Unknown

…

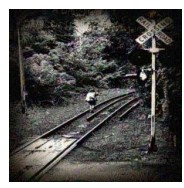

👤 How would you title this image in an exhibition?

🔹 Here are some title ideas for the image, playing on different themes:

**Mysterious:**

* The Crossing

* The Path Less Traveled

* Into the Unknown

* Shadows on the Tracks

* Where the Tracks Lead

…

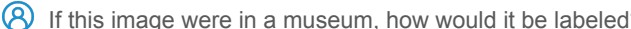

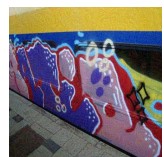

⊗ If this image were in a museum, how would it be labeled?

✦ The image you sent me shows a colorful graffiti mural on the side of a subway train car. If this image were in a museum, it could be labeled in a few different ways, depending on the museum's curatorial perspective and the specific context of the exhibit. Here are a few possibilities:

- **"Graffiti on a train"** This is a straightforward and neutral label that simply identifies the subject of the artwork.
- **"Untitled (Subway car)"** This label is more ambiguous and avoids assigning any particular meaning to the graffiti.
- **"New York Graffiti"** If the graffiti is from New York City, this label would identify its origin and place it within the history of subway graffiti in that city.
- **"Street Art"** This label would emphasize the graffiti's origins outside of the traditional art world.
- **"Artist unknown"** Since graffiti is often anonymous, this label acknowledges that the artist is unknown.

Ultimately, how the graffiti would be labeled would depend on the specific curatorial choices made by the museum.

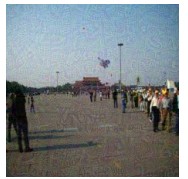

⊗ How would you classify this image in a database?

✦ This image could be classified in a database as follows:

**Category:** Travel, Architecture, Landmarks, China

…

**Location:** Beijing, China

**Date:** Unknown

**Photographer:** Unknown

**Source:** Unknown

**License:** Unknown

**File Size:** Unknown

**Resolution:** Unknown

**Format:** JPEG

**Metadata:** Unknown

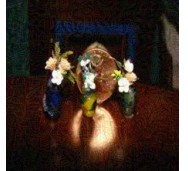

⊗ If this image were the cover of a book, what would its title be?

✦ The image shows a dark room with a wooden table in the center. On the table, there are three vases with white flowers. The vases are all different shapes and colors. The middle vase is the largest and is made of clear glass with a yellow-gold tint. It holds a bouquet of white flowers with light green stems. The flowers are arranged in a symmetrical fashion. The other two vases are smaller and are made of dark blue glass. They hold single white flowers with long green stems. The flowers are placed in the center of the table. There is a copper-colored plate behind the vases in the center of the table. It is reflecting the light from an unknown source. The table appears to be made of a dark wood with a smooth surface. The image is taken in a dark room, and there is a slight shadow behind the table. This suggests that the flowers are the focal point of the image. The light is being directed towards the flowers, highlighting their delicate beauty. The dark background and the arrangement of the flowers create a sense of peace and tranquility. The image could be interpreted as a still life, symbolizing beauty, nature, and harmony.

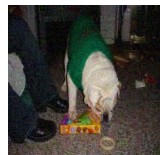 If this image could speak, what would it say?

If this image could speak, it would likely convey a sense of curiosity, exploration, and playfulness. The image shows a person in a green shirt kneeling down and reaching for a small box on the floor. The person's actions suggest that they are interested in the box and want to investigate what's inside. This scene could represent a moment of discovery or curiosity, where the person is drawn to explore the unknown contents of the box.

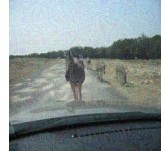 If this image could speak, what would it say?

The image portrays a woman walking through a shallow river, with the water reaching up to her knees. She appears to be alone in the vast landscape, surrounded by nature. The scene is depicted in a artistic manner, with a patterned design overlaying the image, further emphasizing the abstract and artistic nature of the scene. The narrative of the picture could be about self-discovery, the journey of the individual, and the challenges one faces when venturing into unknown territories. The woman's solitary walk through the river might symbolize the search for inner peace, personal growth, or overcoming fears and obstacles in life.

**InternVL**

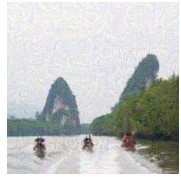 If this image were a song title, what would it be?

If this image were a song title, it could be "River of Mysteries: Journeys to the Unknown Peaks." The title captures the essence of the scene, which features a serene river with towering karst mountains in the background, and people navigating the waters. The title evokes a sense of adventure, exploration, and the unknown, which are central themes in the image.

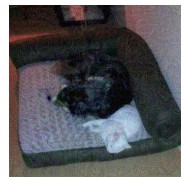 If this were a postcard, what message would it convey?

Based solely on the image, if it were a postcard, the message could convey a sense of mystery or intrigue. The dark, shadowy figure in the center, combined with the eerie glow and the obscured details, might suggest a story of the unknown or the supernatural. The postcard could be used to evoke curiosity or to share a moment of suspense, perhaps as part of a narrative or a thematic collection of images. The overall mood is one of enigma, inviting the viewer to ponder the story behind the scene.

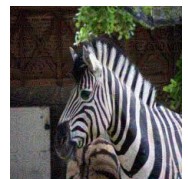 If this image were a chapter in a book, what would its title be?

Based on the image, a possible title for this chapter could be "The Zebra's Secret." The title suggests a narrative that involves the zebra, perhaps revealing something hidden or unknown about the animal.

Table 5: **DynVLA can improve CLIPScore between output text and target text under several state-of-the-art MLLMs.** Larger CLIPScore means closer semantic similarities.

| | Prompt types | Classification | Captioning | General VQA | Specific VQA |
|---|---|---|---|---|---|
| Qwen-VL | Baseline | 0.4527 | 0.1938 | **0.5602** | 0.5637 |
| | DynVLA | **0.4612** | **0.1989** | 0.5596 | **0.5641** |
| InternVL | Baseline | **0.4749** | 0.2059 | 0.4420 | 0.4996 |
| | DynVLA | 0.4719 | **0.2083** | **0.4464** | **0.5001** |
| Gemini | Baseline | 0.5615 | 0.2363 | 0.5257 | 0.5539 |
| | DynVLA | **0.5617** | **0.2393** | **0.5262** | **0.5539** |

Table 6: MLLMs to be used in this work. We list their parameter size, specific components of the language model, vision model, and the vision-language (V-L) connector in the table.

| Model | Parameters | Vision Model | V-L Connector | LLM Scales |
|---|---|---|---|---|
| MiniGPT4 (Zhu et al., 2023) | 8B, 14B | EVA-CLIP-ViT-G | QFormer&Linear | Vicuna-7B&13B, LLaMA2-Chat-7B |
| LLaVA (Liu et al., 2024) | 7.2B, 13.4B | OpenAI-CLIP-ViT-L | Linear/MLP | Vicuna-v0-7B&13B, LLaMA2-Chat-13B, LLaMA-v1.5-7B&13B |
| BLIP2 (Li et al., 2023) | 3B, 8B, 4B, 12B | EVA-CLIP-ViT-G | QFormer | Opt2.7B&6.7B, FlanT5-XL&XXL |
| InstructBLIP (Dai et al., 2024) | 8B, 14B, 4B, 12B | EVA-CLIP-ViT-G | QFormer | Vicuna-v0-7B&13B, FlanT5-XL&XXL |
| Qwen-VL-Chat (Bai et al., 2023) | 9.6B | OpenCLIP-CLIP-ViT-bigG | CrossAttn | Qwen-7B |
| InternVL2 (Chen et al., 2024) | 8B | InternViT | MLP | InternLM-2.5 |
| Google Gemini (Team et al., keyword) | N/A | N/A | N/A | N/A |

## C PROMPTS FOR DIFFERENT TASKS

**Prompts for VQA**
*Any cutlery items visible in the image?*
*Any bicycles visible in this image?*
*Any boats visible in the image?*
*Any bottles present in the image?*
*Are curtains noticeable in the image?*
*Are flags present in the image?*
*Are flowers present in the image?*
*Are fruits present in the image?*
*Are glasses discernible in the image?*
*Are hills visible in the image?*
*Are plates discernible in the image?*
*Are shoes visible in this image?*
*Are there any insects in the image?*
*Are there any ladders in the image?*
*Are there any man-made structures in the image?*
*Are there any signs or markings in the image?*
*Are there any street signs in the image?*
*Are there balloons in the image?*
*Are there bridges in the image?*
*Are there musical notes in the image?*
*Are there people sitting in the image?*
*Are there skyscrapers in the image?*
*Are there toys in the image?*
*Are toys present in this image?*
*Are umbrellas discernible in the image?*
*Are windows visible in the image?*
*Can birds be seen in this image?*
*Can stars be seen in this image?*
*Can we find any bags in this image?*
*Can you find a crowd in the image?*
*Can you find a hat in the image?*
*Can you find any musical instruments in this image?*

*Can you identify a clock in this image?*
*Can you identify a computer in this image?*
*Can you see a beach in the image?*
*Can you see a bus in the image?*
*Can you see a mailbox in the image?*
*Can you see a mountain in the image?*
*Can you see a staircase in the image?*
*Can you see a stove or oven in the image?*
*Can you see a sunset in the image?*
*Can you see any cups or mugs in the image?*
*Can you see any jewelry in the image?*
*Can you see shadows in the image?*
*Can you see the sky in the image?*
*Can you spot a candle in this image?*
*Can you spot a farm in this image?*
*Can you spot a pair of shoes in the image?*
*Can you spot a rug or carpet in the image?*
*Can you spot any dogs in the image?*
*Can you spot any snow in the image?*
*Do you notice a bicycle in the image?*
*Does a ball feature in this image?*
*Does a bridge appear in the image?*
*Does a cat appear in the image?*
*Does a fence appear in the image?*
*Does a fire feature in this image?*
*Does a mirror feature in this image?*
*Does a table feature in this image?*
*Does it appear to be nighttime in the image?*
*Does it look like an outdoor image?*
*Does it seem to be countryside in the image?*
*Does the image appear to be a cartoon or comic strip?*
*Does the image contain any books?*
*Does the image contain any electronic devices?*
*Does the image depict a road?*
*Does the image display a river?*
*Does the image display any towers?*
*Does the image feature any art pieces?*
*Does the image have a lamp?*
*Does the image have any pillows?*
*Does the image have any vehicles?*
*Does the image have furniture?*
*Does the image primarily display natural elements?*
*Does the image seem like it was taken during the day?*
*Does the image seem to be taken indoors?*
*Does the image show any airplanes?*
*Does the image show any benches?*
*Does the image show any landscapes?*
*Does the image show any movement?*
*Does the image show any sculptures?*
*Does the image show any signs?*
*Does the image show food?*
*Does the image showcase a building?*
*How many animals are present in the image?*
*How many bikes are present in the image?*
*How many birds are visible in the image?*
*How many buildings can be identified in the image?*
*How many cars can be seen in the image?*
*How many doors can you spot in the image?*
*How many flowers can be identified in the image?*

972  *How many trees feature in the image?*
973  *Is a chair noticeable in the image?*
974  *Is a computer visible in the image?*
975  *Is a forest noticeable in the image?*
976  *Is a painting visible in the image?*
977  *Is a path or trail visible in the image?*
978  *Is a phone discernible in the image?*
979  *Is a train noticeable in the image?*
980  *Is sand visible in the image?*
981  *Is the image displaying any clouds?*
982  *Is the image set in a city environment?*
983  *Is there a plant in the image?*
984  *Is there a source of light visible in the image?*
985  *Is there a television displayed in the image?*
986  *Is there grass in the image?*
987  *Is there text in the image?*
988  *Is water visible in the image, like a sea, lake, or river?*
989  *How many people are captured in the image?*
990  *How many windows can you count in the image?*
991  *How many animals, other than birds, are present?*
992  *How many statues or monuments stand prominently in the scene?*
993  *How many streetlights are visible?*
994  *How many items of clothing can you identify?*
995  *How many shoes can be seen in the image?*
996  *How many clouds appear in the sky?*
997  *How many pathways or trails are evident?*
998  *How many bridges can you spot?*
999  *How many boats are present, if it's a waterscape?*
1000 *How many pieces of fruit can you identify?*
1001 *How many hats are being worn by people?*
1002 *How many different textures can you discern?*
1003 *How many signs or billboards are visible?*
1004 *How many musical instruments can be seen?*
1005 *How many flags are present in the image?*
1006 *How many mountains or hills can you identify?*
1007 *How many books are visible, if any?*
1008 *How many bodies of water, like ponds or pools, are in the scene?*
1009 *How many shadows can you spot?*
1010 *How many handheld devices, like phones, are present?*
1011 *How many pieces of jewelry can be identified?*
1012 *How many reflections, perhaps in mirrors or water, are evident?*
1013 *How many pieces of artwork or sculptures can you see?*
1014 *How many staircases or steps are in the image?*
1015 *How many archways or tunnels can be counted?*
1016 *How many tools or equipment are visible?*
1017 *How many modes of transportation, other than cars and bikes, can you spot?*
1018 *How many lamp posts or light sources are there?*
1019 *How many plants, other than trees and flowers, feature in the scene?*
1020 *How many fences or barriers can be seen?*
1021 *How many chairs or seating arrangements can you identify?*
1022 *How many different patterns or motifs are evident in clothing or objects?*
1023 *How many dishes or food items are visible on a table setting?*
1024 *How many glasses or mugs can you spot?*
1025 *How many pets or domestic animals are in the scene?*
     *How many electronic gadgets can be counted?*
     *Where is the brightest point in the image?*
     *Where are the darkest areas located?*
     *Where can one find leading lines directing the viewer's eyes?*
     *Where is the visual center of gravity in the image?*

1026 *Where are the primary and secondary subjects positioned?*
1027 *Where do the most vibrant colors appear?*
1028 *Where is the most contrasting part of the image located?*
1029 *Where does the image place emphasis through scale or size?*
1030 *Where do the textures in the image change or transition?*
1031 *Where does the image break traditional compositional rules?*
1032 *Where do you see repetition or patterns emerging?*
1033 *Where does the image exhibit depth or layers?*
1034 *Where are the boundary lines or borders in the image?*
1035 *Where do different elements in the image intersect or overlap?*
1036 *Where does the image hint at motion or movement?*
1037 *Where are the calm or restful areas of the image?*
1038 *Where does the image become abstract or less defined?*
1039 *Where do you see reflections, be it in water, glass, or other surfaces?*
1040 *Where does the image provide contextual clues about its setting?*
1041 *Where are the most detailed parts of the image?*
1042 *Where do you see shadows, and how do they impact the composition?*
1043 *Where can you identify different geometric shapes?*
1044 *Where does the image appear to have been cropped or framed intentionally?*
1045 *Where do you see harmony or unity among the elements?*
1046 *Where are there disruptions or interruptions in patterns?*
1047 *What is the spacing between objects or subjects in the image?*
1048 *What foreground, mid-ground, and background elements can be differentiated?*
1049 *What type of energy or vibe does the image exude?*
1050 *What might be the sound environment based on the image's content?*
1051 *What abstract ideas or concepts does the image seem to touch upon?*
1052 *What is the relationship between the main subjects in the image?*
1053 *What items in the image could be considered rare or unique?*
1054 *What is the gradient or transition of colors like in the image?*
1055 *What might be the smell or aroma based on the image's content?*
1056 *What type of textures can be felt if one could touch the image's content?*
1057 *What boundaries or limits are depicted in the image?*
1058 *What is the socioeconomic context implied by the image?*
1059 *What might be the immediate aftermath of the scene in the image?*
1060 *What seems to be the main source of tension or harmony in the image?*
1061 *What might be the narrative or backstory of the main subject?*
1062 *What elements of the image give it its primary visual weight?*
1063 *Would you describe the image as bright or dark?*
1064 *Would you describe the image as colorful or dull?*

**Prompts for Image Classification**
*Identify the primary theme of this image in one word.*
*How would you label this image with a single descriptor?*
*Determine the main category for this image.*
*Offer a one-word identifier for this picture.*
*If this image were a file on your computer, what would its name be?*
*Tag this image with its most relevant keyword.*
*Provide the primary classification for this photograph.*
*How would you succinctly categorize this image?*
*Offer the primary descriptor for the content of this image.*
*If this image were a product, what label would you place on its box?*
*Choose a single word that encapsulates the image's content.*
*How would you classify this image in a database?*
*In one word, describe the essence of this image.*
*Provide the most fitting category for this image.*
*What is the principal subject of this image?*
*If this image were in a store, which aisle would it belong to?*

*Provide a singular term that characterizes this picture.*
*How would you caption this image in a photo contest?*
*Select a label that fits the main theme of this image.*
*Offer the most appropriate tag for this image.*
*Which keyword best summarizes this image?*
*How would you title this image in an exhibition?*
*Provide a succinct identifier for the image's content.*
*Choose a word that best groups this image with others like it.*
*If this image were in a museum, how would it be labeled?*
*Assign a central theme to this image in one word.*
*Tag this photograph with its primary descriptor.*
*What is the overriding theme of this picture?*
*Provide a classification term for this image.*
*How would you sort this image in a collection?*
*Identify the main subject of this image concisely.*
*If this image were a magazine cover, what would its title be?*
*What term would you use to catalog this image?*
*Classify this picture with a singular term.*
*If this image were a chapter in a book, what would its title be?*
*Select the most fitting classification for this image.*
*Define the essence of this image in one word.*
*How would you label this image for easy retrieval?*
*Determine the core theme of this photograph.*
*In a word, encapsulate the main subject of this image.*
*If this image were an art piece, how would it be labeled in a gallery?*
*Provide the most concise descriptor for this picture.*
*How would you name this image in a photo archive?*
*Choose a word that defines the image's main content.*
*What would be the header for this image in a catalog?*
*Classify the primary essence of this picture.*
*What label would best fit this image in a slideshow?*
*Determine the dominant category for this photograph.*
*Offer the core descriptor for this image.*
*If this image were in a textbook, how would it be labeled in the index?*
*Select the keyword that best defines this image's theme.*
*Provide a classification label for this image.*
*If this image were a song title, what would it be?*
*Identify the main genre of this picture.*
*Assign the most apt category to this image.*
*Describe the overarching theme of this image in one word.*
*What descriptor would you use for this image in a portfolio?*
*Summarize the image's content with a single identifier.*
*Imagine you're explaining this image to someone over the phone. Please describe the image in one word?*
*Perform the image classification task on this image. Give the label in one word.*
*Imagine a child is trying to identify the image. What might they excitedly point to and name?*
*If this image were turned into a jigsaw puzzle, what would the box label say to describe the picture inside?*
*Classify the content of this image.*
*If you were to label this image, what label would you give?*
*What category best describes this image?*
*Describe the central subject of this image in a single word.*
*Provide a classification for the object depicted in this image.*
*If this image were in a photo album, what would its label be?*
*Categorize the content of the image.*
*If you were to sort this image into a category, which one would it be?*
*What keyword would you associate with this image?*
*Assign a relevant classification to this image.*
*If this image were in a gallery, under which section would it belong?*

*Describe the main theme of this image in one word.*
*Under which category would this image be cataloged in a library?*
*What classification tag fits this image the best?*
*Provide a one-word description of this image's content.*
*If you were to archive this image, what descriptor would you use?*

**Prompts for Image Captioning**
*Elaborate on the elements present in this image.*
*In one sentence, summarize the activity in this image.*
*Relate the main components of this picture in words.*
*What narrative unfolds in this image?*
*Break down the main subjects of this photo.*
*Give an account of the main scene in this image.*
*In a few words, state what this image represents.*
*Describe the setting or location captured in this photograph.*
*Provide an overview of the subjects or objects seen in this picture.*
*Identify the primary focus or point of interest in this image.*
*What would be the perfect title for this image?*
*How would you introduce this image in a presentation?*
*Present a quick rundown of the image's main subject.*
*What's the key event or subject captured in this photograph?*
*Relate the actions or events taking place in this image.*
*Convey the content of this photograph in a single phrase.*
*Offer a succinct description of this picture.*
*Give a concise overview of this image.*
*Translate the contents of this picture into a sentence.*
*Describe the characters or subjects seen in this image.*
*Capture the activities happening in this image with words.*
*How would you introduce this image to an audience?*
*State the primary events or subjects in this picture.*
*What are the main elements in this photograph?*
*Provide an interpretation of this image's main event or subject.*
*How would you title this image for an art gallery?*
*What scenario or setting is depicted in this image?*
*Concisely state the main actions occurring in this image.*
*Offer a short summary of this photograph's contents.*
*How would you annotate this image in an album?*
*If you were to describe this image on the radio, how would you do it?*
*In your own words, narrate the main event in this image.*
*What are the notable features of this image?*
*Break down the story this image is trying to tell.*
*Describe the environment or backdrop in this photograph.*
*How would you label this image in a catalog?*
*Convey the main theme of this picture succinctly.*
*Characterize the primary event or action in this image.*
*Provide a concise depiction of this photo's content.*
*Write a brief overview of what's taking place in this image.*
*Illustrate the main theme of this image with words.*
*How would you describe this image in a gallery exhibit?*
*Highlight the central subjects or actions in this image.*
*Offer a brief narrative of the events in this photograph.*
*Translate the activities in this image into a brief sentence.*
*Give a quick rundown of the primary subjects in this image.*
*Provide a quick summary of the scene captured in this photo.*
*How would you explain this image to a child?*
*What are the dominant subjects or objects in this photograph?*
*Summarize the main events or actions in this image.*

*Describe the context or setting of this image briefly.*
*Offer a short description of the subjects present in this image.*
*Detail the main scenario or setting seen in this picture.*
*Describe the main activities or events unfolding in this image.*
*Provide a concise explanation of the content in this image.*
*If this image were in a textbook, how would it be captioned?*
*Provide a summary of the primary focus of this image.*
*State the narrative or story portrayed in this picture.*
*How would you introduce this image in a documentary?*
*Detail the subjects or events captured in this image.*
*Offer a brief account of the scenario depicted in this photograph.*
*State the main elements present in this image concisely.*
*Describe the actions or events happening in this picture.*
*Provide a snapshot description of this image's content.*
*How would you briefly describe this image's main subject or event?*
*Describe the content of this image.*
*What's happening in this image?*
*Provide a brief caption for this image.*
*Tell a story about this image in one sentence.*
*If this image could speak, what would it say?*
*Summarize the scenario depicted in this image.*
*What is the central theme or event shown in the picture?*
*Create a headline for this image.*
*Explain the scene captured in this image.*
*If this were a postcard, what message would it convey?*
*Narrate the visual elements present in this image.*
*Give a short title to this image.*
*How would you describe this image to someone who can't see it?*
*Detail the primary action or subject in the photo.*
*If this image were the cover of a book, what would its title be?*
*Translate the emotion or event of this image into words.*
*Compose a one-liner describing this image's content.*
*Imagine this image in a magazine. What caption would go with it?*
*Capture the essence of this image in a brief description.*
*Narrate the visual story displayed in this photograph.*

