# OpenReview forum: "Improving Adversarial Transferability in MLLMs via Dynamic Vision-Language Alignment Attack"
_ICLR.cc/2025/Conference — ICLR 2025 Conference Withdrawn Submission_

### Official Review · Reviewer_np2J · 2024-11-02

**Soundness:** 3
**Presentation:** 2
**Contribution:** 2
**Rating:** 5
**Confidence:** 3

**Summary:**

Multimodal Large Language Models (MLLMs) are susceptible to adversarial attacks, but the transferability of these attacks across different models is still limited. While current methods primarily target vision-specific perturbations, this paper focuses on attacking the vision-language modality alignment. Specifically, the authors propose DynVLA. a novel approach that injects dynamic perturbations into the vision-language connector. Experimental results demonstrate that DynVLA effectively transfers adversarial examples across Q-Former models.

**Strengths:**

1.	This paper studies the transferability of adversarial examples in attacking MLLMs, which is important and practical.
2.	This paper proposes a novel approach, DynVLA, which generates adversarial examples by targeting vision-language alignment rather than solely vision-specific perturbations.
3.	The experiments with both open- and closed-source models and different tasks (i.e., classification, captioning, VQA) provides a comprehensive evaluation of the method’s applicability.

**Weaknesses:**

1.	The authors' main claim about the transferability of DynVLA is not well-supported. According to the experimental results, the adversarial examples are mainly effective for models with the Q-former architecture (i.e., BLIP2, InstructBLIP, and MiniGPT4), but show limited success with other architectures (i.e., LLaVA and Gemini). This undermines the paper's key contribution regarding the transferability of adversarial examples.
2.	This paper does not convincingly demonstrate that DynVLA primarily targets vision-language modality alignment. The adversarial examples produced by DynVLA also include perturbations on the original image, which could potentially attack the vision encoder rather than the vision-language alignment in MLLMs. The authors could conduct additional experiments to verify this. For instance, they could randomly generate noise with the same bit size as the noise created by DynVLA and compare the ASR of these two perturbations.
3.	The experiments do not compare with the latest and most advanced baselines. As mentioned in the paper, both [1] and [2] have explored transferring adversarial examples across different models. However, three of the four baselines in this study were published before 2020. Why did the authors select these older baselines while overlooking the latest methods in this field? They should include more empirical comparisons with these existing works.

Reference

[1] Yunqing Zhao, Tianyu Pang, Chao Du, Xiao Yang, Chongxuan Li, Ngai-Man Man Cheung, and Min Lin. On evaluating adversarial robustness of large vision-language models. Advances in Neural Information Processing Systems, 36, 2024.

[2] Haochen Luo, Jindong Gu, Fengyuan Liu, and Philip Torr. An image is worth 1000 lies: Adversarial transferability across prompts on vision-language models. arXiv preprint arXiv:2403.09766, 2024.

**Questions:**

1.	There are 4 baselines in the experiments. What does “baseline” refer in Table 1,2 and 3?

---

> ### Author Response · Authors · 2024-11-15
>
> **Q1. DynVLA is mainly effective for models with the Q-former architecture.**
>
> **A:** Our method is not limited to Q-former architecture. Besides the transferability results on BLIP2, InstructBLIP and MiniGPT4, we also show some attacking success cases on models with very different structures like Qwen-VL, InternVL and even closed-source model Gemini(in Appendix). Our method can make the output of Gemini contain our target texts, while the baseline method can’t.
> DynVLA can also demonstrate a clear improvement over the baseline in LLaVA models. In some cases, our method’s ASR is doubled compared to the baseline method.
>
> **Q2. Does not convincingly demonstrate that DynVLA primarily targets vision-language modality alignment**
>
> **A:** The purpose of adding noise is to cover diverse vision-language modality alignment cases and will only be used when crafting adversarial examples. So the noise used by DynVLA is not as same as the adversarial perturbations. When evalutating, the image with the adversarial perturbation will be the only input. The reviewer mentioned that “The adversarial examples produced by DynVLA also include perturbations on the original image, which could potentially attack the vision encoder rather than the vision-language alignment in MLLMs”, the generated adversarial example should attack the whole MLLMs instead of only changing the sample’s vision-language alignment.
>
> **Q3. Doesn’t compare with the latest and most advanced baselines**
>
> **A:** We discuss these two papers and other more advanced transfer attack works on MLLMs in our Introduction and Related Works. However, most of these works are not under targeted text attack settings, for example, [1] makes the adversarial example’s visual representation close to another image (or images generated by generative models) and only attacks the vision encoder of MLLMs, thus can’t be used directly to craft adversarial examples for target text output. Our method attacks the whole MLLM end-to-end. [2] focus on the transferability across prompts, and they also indicated that the cross-model transferability is very low in their paper and leave this for further work. Our method is orthogonal with this paper’s approach and hopefully can be combined with their method to craft transferable adversarial examples across both prompts and models. We choose some traditional transfer attack as our baseline as they can be used under our settings directly, most of them are input-transformation based method. SIT[3] we used as baseline combines a set of input-transformations and is the best traditional transfer attack method under our setting.
>
> **Q4:** the baseline is the PGD attack without any transfer attack method.
>
> [1] Yunqing Zhao, Tianyu Pang, Chao Du, Xiao Yang, Chongxuan Li, Ngai-Man Man Cheung, and Min Lin. On evaluating adversarial robustness of large vision-language models. Advances in Neural Information Processing Systems, 36, 2024.
>
> [2] Haochen Luo, Jindong Gu, Fengyuan Liu, and Philip Torr. An image is worth 1000 lies: Adversarial transferability across prompts on vision-language models. arXiv preprint arXiv:2403.09766, 2024.
>
> [3] Structure invariant transformation for better adversarial transferability. In Proceedings of the IEEE/CVF International Conference on Computer Vision, pp. 4607–4619, 2023a.

---

### Official Review · Reviewer_sRoA · 2024-11-02

**Soundness:** 3
**Presentation:** 3
**Contribution:** 2
**Rating:** 3
**Confidence:** 5

**Summary:**

This paper introduces a Dynamic Vision-Language Alignment Attack (DynVLA) to enhance the adversarial robustness of Multi-Modal Large Language Models (MLLMs) by addressing the interaction between visual and textual modalities. Existing baseline attack methods typically generate adversarial samples based on a single type of vision-language alignment, leading to low transferability across different models. DynVLA dynamically perturbs the attention mechanism applied to visual tokens to improve interactions between visual and textual information, resulting in more diverse adversarial samples. Specifically, it employs a Gaussian kernel to smoothly adjust the model's attention distribution, allowing for more precise perturbations of specific regions in the visual input. Experimental results demonstrate that DynVLA outperforms traditional methods across multiple benchmark datasets.

**Strengths:**

(1) Introduces a novel attack method, DynVLA, which dynamically adjusts vision-language interactions, improving on traditional adversarial strategies.
(2) Well-founded methodology based on established adversarial techniques. And Gaussian smoothing adds refinement.
(3) Generally clear structure, though some mathematical sections could be improved with more detailed explanations.
(4 ) Addresses a critical gap in MLLM robustness, with potential implications for multi-modal AI security.

**Weaknesses:**

(1) While DynVLA introduces a dynamic perturbation approach, the core methodology remains relatively straightforward and does not offer groundbreaking innovations compared to existing techniques.
Dependency on White-Box Models:
(2) The effectiveness of DynVLA hinges on having access to the internal structure of the model being attacked. The method's applicability diminishes significantly for black-box models or those where internal mechanisms are inaccessible. This limitation restricts the method's utility in real-world scenarios where such access is often unavailable.
(3) The decision to use "unknown" as the targeted output raises concerns regarding its relevance. This output is not representative of typical outputs from MLLMs, making it questionable whether the attack accurately simulates realistic adversarial conditions. A more grounded target could enhance the validity of the results and provide clearer insights into the method's practical effectiveness.
(4) The experiments primarily focus on specific datasets, which may not reflect the broader applicability of the method. Expanding the experimental framework to include diverse datasets and model architectures would strengthen the findings and validate the robustness of the approach.

**Questions:**

See weakness.

---

> ### Author Response · Authors · 2024-11-15
>
> **Q1. The core methodology remains relatively straightforward and does not offer groundbreaking innovations compared to existing techniques**
>
> **A:** Our work addresses the challenge that targeted attack is hard to transfer between models, and demonstrates that the vision-language modality alignment should be considered under this setting instead of the visual information only, which is ignored by previous works.
> Our experimental results also show that most existing methods of transfer attack are less effective under MLLMs, and other works about transfer attacks on MLLMs mainly focus on misleading the visual concept in the image.
>
>
> **Q2. The effectiveness of DynVLA hinges on having access to the internal structure of the model**
>
> **A:** Without accessing any internal structure, our method can have an improvement compared to the baseline method, and with a similar model structure, the performance of our method will be better. We also show some attacking success cases on closed-source models like Gemini. As our best knowledge, there has been no attack success case under this targeted attack setting before, and attacking these models is still challenged for several reasons, for example, most of state-of-the-art models support dynamic visual input resolutions, this means that the visual input will be split and resized, which may decrease the transferability of adversarial examples.
>
> **Q3. The chosen target output “unknown” is not representative of typical outputs from MLLMs**
>
> **A:** We provide results obtained using different target outputs in Section 4.5 and Figure 3, our DynVLA can improve the transferability on sentences like “I’m sorry”, “I don’t know”, as well as a common object “cat”.  We choose these target outputs as what previous work[1] does. We deliberately choose a not typical output of MLLMs because this is the most difficult case to make MLLMs generate this target output, thus showing the effectiveness of our DynVLA method.
>
> **Q4. The experiments primarily focus on specific datasets**
>
> **A:** Our experimental settings follow the previously published work’s[1] setting. We agree that providing results on more datasets will make our results stronger. Instead of doing more experiments on different datasets, we conduct experiments on different prompt types and show the results in Section 4.4, which demonstrate that our method can be generalized to other task prompts. What’s more, generating 1000 adversarial examples needs ~48 GPU hours on an Nvidia A100 GPU, evaluating on another dataset has a huge cost.
>
>
> [1] An image is worth 1000 lies: Adversarial transferability across prompts on vision-language models, ICLR 2024.

---

### Official Review · Reviewer_1ytZ · 2024-11-04

**Soundness:** 2
**Presentation:** 3
**Contribution:** 2
**Rating:** 3
**Confidence:** 5

**Summary:**

This paper proposes DynVLA, an adversarial attack method that adds Gaussian kernels to the attention map of vision-language connectors, and evaluates it on sufficient models.

**Strengths:**

1. The paper was evaluated on many models, including BLIP-2, InstructBLIP, MiniGPT, and LLaVa.

2. The writing of the paper is clear.

**Weaknesses:**

1. Line 102 says "DynVLA does not require any prior knowledge of the model", I think this sentence is overblown. In Tables 1, 2, and 3, the authors all perform transfer attacks on LVLMs with very similar structures, including BLIP-2, InstructBLIP, and MiniGPT. Their vision-language connectors are all Q-formers. this can hardly be claimed to be without any a priori knowledge of the model. If indeed no a priori knowledge of any model is required, the authors should use their method to attack structurally different models, such as GPT-4, etc., or at least LLaVA.

2. The results are poor. The authors perform transfer attacks in Tables 1, 2 and 3 only in a few LVLMs that use Q-former as a vision-language connector, including BLIP-2, InstructBLIP, and MiniGPT. There seems to be a large improvement from the table (which will be continued to the question in point 3), but once transfer attacks are performed on the other models, such as the LLaVA in Table 4, the performance improvement is only 1%, and even 0.1% on V1.6-LLaMA3 (the authors chose 1000 images as dataset, that is only 1 successful attack).

3. The improvements in Tables 1-3 may be unfair. It is well known that adversarial attacks are very easy to overfit, leading to poorer transferability. The authors used a PGD attack with an extremely large T = 2000 steps (line 296). This leads to severe overfitting, which leads to poor attack transferability. In a way, the authors' approach is more like a regularisation that prevents overfitting, so why not compare it with some similar approaches, and some of the already existing methods for improving the transferability of adversarial attacks？Such as MI [1], DI [2], SGA [3], CroPA [4], etc.

[1] Boosting adversarial attacks with momentum, CVPR 2018.
[2] Improving transferability of adversarial examples with input diversity, CVPR 2019.
[3] Set-level guidance attack: Boosting adversarial transferability of vision-language pre-training models, ICCV 2023.
[4] An image is worth 1000 lies: Adversarial transferability across prompts on vision-language models, ICLR 2024.

4. Overstatement. line 458-460 says "We evaluate the adversarial examples generated by InstructBLIP models on Gemini, ......, these samples on closed-source models like Gemini have never been reported by other works." However, Attack-Bard [5] successfully attacked Google-Bard (a previous version of Gemini) by using a series of boosts to adversarial transferability on models such as InstructBLIP. It also works much better than the author's method.

[5] How Robust is Google’s Bard to Adversarial Image Attacks? NeurIPS 2023 Workshop on Robustness of Few-shot and Zero-shot Learning in Foundation Models

5. Lack of necessary ablation experiments. The authors should have at least demonstrated the adversarial transferability in a conventional PGD attack setup (not T=2000 steps) and compared it with classical adversarial attack methods. The existing ablation experiments are not sufficient.

**Questions:**

See weaknesses.

---

> ### Author Response · Authors · 2024-11-15
>
> We appreciate the reviewer's comments and want to answer some questions the reviewer is concerned about.
>
> **Q1. Our method needs prior structure knowledge of MLLMs.**
>
> **A:** Besides the transferability results on BLIP2, InstructBLIP and MiniGPT4, we also show some attacking success cases on models with very different structures like **Qwen-VL**, **InternVL** and even closed-source model **Gemini**(in Appendix). Our method can make the output of Gemini contain our target texts, while the baseline method can’t. Most of these state-of-the-art models support dynamic visual input resolutions so will also do splitting and upsampling operations when preprocessing the visual input, which may decrease the transferability of the adversarial examples.
>
> We didn’t show the transfer attack results between LLaVA and InstructBLIP due to their **different visual input resolutions**. So with the same reason mentioned, the adversarial example generated by InstructBLIP should be resized to a high resolution for LLaVA to evaluate. The resize operation will somehow make the adversarial examples not effective enough.
>
> It is also important that BLIP2, InstructBLIP and MiniGPT4 have similar structures in their vision-language connectors, but the based LLMs vary from OPT, FlanT5 to Vicuna, these base LLMs’ structures are quite different, our method can generate highly transferable adversarial examples on different language spaces. Transferring between MLLMs with completely different architectures is much harder is also observed by [3], we discussed this in our limitation section.
>
> **Q2. The results on LLaVA models are poor.**
>
> **A:** The performance of both the baseline and our DynVLA method is relatively low on LLaVA; however, DynVLA demonstrates a clear improvement over the baseline. In some cases, our method’s ASR is doubled compared to the baseline method. In the case of LLaVA-V1.6-LLaMA3, the baseline method fails to achieve any successful attacks, whereas our method successfully generates effective adversarial examples.
>
> **Q3&Q5. The baseline method is overfitting so have low performance.**
>
> **A:** This a good question, we conduct new ablation experiments and the results show that the baseline method does have a slight overfitting when the attack steps T = 2000. However, the drop of ASR between T=2000 and the best ASR is very small compared to the improvement of our method.
>
> We follow the previous work’s[1] setting to set the attack steps to 2000. The optimization of MLLM is complex so conventional PGD attack setup with smaller attack steps may be ineffective. What’s more, some individual samples need more steps to be attacked than others under wight-box settings, so we choose T = 2000 for a more fair comparison.
>
> We have compared DynVLA with other methods and show the results in Section 4.3 and Figure 2. We compared with [5] and discussed why [4] is not effective under MLLM and finally decided to not show it in our results(its performance is worse than the PGD baseline). [6] focus on attacking Vision-Language Pretrained Models so can’t be used under our settings directly. [1] focus on the transferability across prompts, and they also indicated that the cross-model transferability is very low in their paper and leave this for further work. Our method is orthogonal with this paper’s approach and hopefully can be combined with their method to craft transferable adversarial examples across both prompts and models.
>
> |     steps     |   100 |   200 |   400 |   600 |   800 |   1000 |   1200 |   1400 |   1600 |   1800 |   2000 |
> |:---------|------:|------:|------:|------:|------:|-------:|-------:|-------:|-------:|-------:|-------:|
> | baseline | 0.127 | 0.194 | 0.22  | 0.218 | 0.218 |  0.211 |  0.217 |  0.211 |  0.212 |  0.214 |  0.21  |
> | ours     | 0.062 | 0.269 | 0.396 | 0.489 | 0.527 |  0.546 |  0.569 |  0.579 |  0.592 |  0.596 |  0.596 |
>
> **Q4. Previous work[2] had successful attacking Gemini(Google-Bard) case and had better performance than us**
>
> **A:** The difference between this paper and our paper is: this paper focuses on **untargeted** attack, which means the attack will be considered as success if the **description of the vision concept** isn’t generated by MLLMs. So Attack-Bard conducted the attack on the **visual encoder of the MLLM**. However, our work mainly focuses on the **targeted** attack, the attack will be considered as success only when the output text **exactly matches the targeted text**, which is much harder than the untargeted setting and has to perform an end-to-end attack approach, we claimed that the attack success case **under our setting** has never been reported before. And comparing the ASR between our method and Attack-Bard directly is unfair.

---

> > ### Author Response · Authors · 2024-11-15
> >
> > [1] An image is worth 1000 lies: Adversarial transferability across prompts on vision-language models, ICLR 2024.
> >
> > [2] How Robust is Google’s Bard to Adversarial Image Attacks? NeurIPS 2023 Workshop on Robustness of Few-shot and Zero-shot Learning in Foundation Models
> >
> > [3] When do universal image jailbreaks transfer between vision-language models? arXiv preprint arXiv:2407.15211, 2024.
> >
> > [4] Boosting adversarial attacks with momentum, CVPR 2018.
> >
> > [5] Improving transferability of adversarial examples with input diversity, CVPR 2019.
> >
> > [6] Set-level guidance attack: Boosting adversarial transferability of vision-language pre-training models, ICCV 2023.

---

### Official Review · Reviewer_7VLz · 2024-11-04

**Soundness:** 2
**Presentation:** 3
**Contribution:** 2
**Rating:** 5
**Confidence:** 3

**Summary:**

This paper proposes a novel approach to inject dynamic perturbations into the vision-language connector to enhance vision-language alignment of different models. It adds a Gaussian Kernel to the attention map to force the model focus on a specific area. The experimental results indicate the effectiveness of the proposed attack.

**Strengths:**

1.The paper is well written.

2. The paper provides comprehensive evaluation on different model architectures, such as BLIP2, MINIGPT4, InstructBLIP, LLaVA associated with different surrogate model.

**Weaknesses:**

1. DynVLA adds a Gaussian kernel to the attention map, will it change the forward process? For example, in a standard forward process, there is no Gaussian kernel. In proposed attack, with the Gaussian kernel, it achieves good attack performance. However, if the users use the provided model, it would be easily observed due to the unusual forward pass. It will hurt the attack’s stealthies.

2. Some analysis regarding the defense mechanism would be good. For example, will the attack resistant to the defense?
3. For the examples given in Figure4, it seems the attack makes the outputs contains word “unknown

**Questions:**

Please check the weakness

---

> ### Author Response · Authors · 2024-11-15
>
> **Q1. Will our method change the forward process?**
>
> **A**: Our method will only be used when crafting adversarial examples using the surrogate models and the adversarial examples will be the input of a black-box model. The inference process of the black-box model will never be changed since we have no access to that model.
>
> **Q2. Defense mechanism**
>
> **A**: Thanks for your suggestion, but an attack that can resist defense is not our main focus.  We will consider add it.
>
> **Q3. Examples in Figure 4 contain the word “unknown”**
>
> **A**: Yes this is what we aim to do. The attacking purpose is to make the model generate target text(“unknown” in this case).

---

### Note · Authors · 2024-11-15

I have read and agree with the venue's withdrawal policy on behalf of myself and my co-authors.